# Comparison of interventions for Barrett's esophagus: A network meta-analysis

Qinlin Zhang[1,2☯], Miya Li[3☯], Xin Jin[3], Ruhong Zhou[3], Yize Ying[2], Xueping Wu[2], Jiyong Jing[4], Wensheng Pan[2,4]*

1 Department of Gastroenterology, Sanmen County People's Hospital, Taizhou, Zhejiang, China, 2 Zhejiang Chinese Medical University, Hangzhou, Zhejiang, China, 3 Wenzhou Medical University, Wenzhou, Zhejiang, China, 4 Department of Gastroenterology, Zhejiang Provincial People's Hospital, People's Hospital of Hangzhou Medical College, Hangzhou, Zhejiang, China

☯ These authors contributed equally to this work.
* wspan223@163.com

**Data Availability Statement:** All relevant data are within the manuscript and its Supporting Information files.

**Funding:** The authors received no specific funding for this work.

## Abstract

### Background and objective

Barrett's esophagus (BE) is a precancerous condition that has the potential to develop into esophageal cancer (EC). Currently, there is a wide range of management options available for individuals at different pathological stages in Barrett's esophagus (BE). However, there is currently a lack of knowledge regarding their comparative efficacy. To address this gap, we conducted a network meta-analysis of published randomized controlled trials to examine the comparative effectiveness of all regimens.

### Methods

Data extracted from eligible randomized controlled trials were utilized in a Bayesian network meta-analysis to examine the relative effectiveness of BE's treatment regimens and determine their ranking in terms of efficacy. The ranking probability for each regimen was assessed using the surfaces under cumulative ranking values. The outcomes under investigation were complete ablation of BE, neoplastic progression of BE, and complete eradication of dysplasia.

### Results

We identified twenty-three RCT studies with a total of 1675 participants, and ten different interventions. Regarding complete ablation of non-dysplastic BE, the comparative effectiveness ranking indicated that argon plasma coagulation (APC) was the most effective regimen, with the highest SUCRA value, while surveillance and PPI/H2RA were found to be the least efficacious regimens. For complete ablation of BE with low-grade dysplasia, high-grade dysplasia, or esophageal cancer, photodynamic therapy (PDT) had the highest SUCRA value of 94.1%, indicating it as the best regimen. Additionally, for complete eradication of dysplasia, SUCRA plots showed a trend in ranking PDT as the highest with a SUCRA value of 91.2%. Finally, for neoplastic progression, radiofrequency ablation (RFA) and surgery were found to perform significantly better than surveillance. The risk of bias

                                                                1 / 16

**Competing interests:** The authors declare that they have no conflict of interest.

**Abbreviations:** BE, Barrett's esophagus; EC, esophageal cancer; APC, argon plasma coagulation; PDT, photodynamic therapy; RFA, radiofrequancy ablation; EMR, endoscopic mucosal resection; NWM, network meta-analysis; RCTs, Randomized controlled trials; NDBE, non-dysplastic BE; LGD, low-grade dysplasia; HGD, high-grade dysplasia; EC, esophageal cancer.

assessment revealed that 6 studies had an overall high risk of bias. However, meta-regression with risk of bias as a covariate did not indicate any influence on the model. In terms of the Confidence in Network Meta-Analysis evaluation, a high level of confidence was found for all treatment comparisons.

## Conclusion

Endoscopic surveillance alone or PPI/H2RA alone may not be sufficient for managing BE, even in cases of non-dysplastic BE. However, APC has shown excellent efficacy in treating non-dysplastic BE. For cases of BE with low-grade dysplasia, high-grade dysplasia, or esophageal cancer, PDT may be the optimal intervention as it can induce regression of BE metaplasia and prevent future progression of BE to dysplasia and EC.

## Introduction

Barrett's esophagus (BE) is considered a precancerous condition that can lead to esophageal cancer (EC). According to a large population-based study, the relative risk of adenocarcinoma was 11 times higher among patients with BE than in the general population [1]. Therefore, it is important to identify optimal interventions that promote regression of BE metaplasia and prevent progression to dysplasia and EC. However, current practices for managing BE vary across the world due to several existing national guidelines [2–4]. Successful ablation of BE is an attractive concept, and there has been considerable enthusiasm for related techniques. Although there are many treatment options for BE, such as drug control, endoscopic surveillance, argon plasma coagulation (APC), photodynamic therapy (PDT), radiofrequency ablation (RFA), endoscopic mucosal resection (EMR), and endoscopic submucosal dissection (ESD), there is currently no clear international consensus on their ranking in terms of curative effects.

Network meta-analysis (NWM) is an evidence synthesis tool that compares randomized controlled trials (RCTs) with multiple treatments [5,6]. NWM incorporates both direct and indirect evidence from a collection of RCTs, providing information on the relative effects of three or more therapeutic interventions competing for the same result.

## Methods

### Identification of studies and data extraction

In this NWM, we followed a systematic process that included four steps: identification, screening, eligibility, and inclusion. We searched the PubMed/MEDLINE, Embase, and Web of Science databases up to March 2023 to identify human studies published in English using the following search terms and/or Medical Subject Headings: ("Barrett Esophagus" OR "Barrett Metaplasia" OR "Barrett Metaplasias" OR "Metaplasia, Barrett" OR "Metaplasias, Barrett" OR "Barrett's Syndrome" OR "Barretts Syndrome" OR "Barrett Syndrome" OR "Barrett's Esophagus" OR "Barretts Esophagus" OR "Esophagus, Barrett's" OR "Esophagus, Barrett" OR "Barrett Epithelium" OR "Epithelium, Barrett") AND (treatment OR therapy OR therapeutics) AND ("Randomized Controlled Trial" OR "Clinical Trials, Randomized" OR "Trials, Randomized Clinical" OR "Controlled Clinical Trials, Randomized" OR "RCTs"). Additionally, we conducted a manual search of all review articles, published editorials, and original studies

retrieved. Two authors independently extracted data from each study, and any disagreements were resolved through discussion with a third author until consensus was reached. We performed this NWM according to the Preferred Reporting Items for Systematic Reviews and Meta-Analyses extension statement for interventions [7], and the quality of evidence derived from pairwise and NWM was evaluated using the GRADE(Grading of Recommendations Assessment, Development and Evaluation) working group modality. Furthermore, we appraised the confidence in estimates derived from NWM.

## Study selection

Only randomized controlled trials (RCTs) treating Barrett esophagus were included in this study. Since there are many different techniques and treatment regimens used for Barrett esophagus, the authors grouped all treatments into 10 major groups based on the selected studies, which are shown in Table 1. To ensure the quality of the included RCTs, we excluded studies with small sample sizes below 20. In addition to this criterion, the studies had to contain at least one of the following three outcomes: complete ablation of BE, neoplastic progression of BE, or complete eradication of dysplasia. Complete ablation of BE is defined as the transformation from Barrett's esophagus to normal esophageal squamous epithelium during endoscopic re-examination. Neoplastic progression of BE was defined as progression of any of the following dysplasia severities: low-grade dysplasia, high-grade dysplasia, or esophageal cancer.

## Data extraction

Two researchers performed this process independently, using a standardized form that included the author's name, year of publication, country, number of participants, age, gender, treatment regimens, duration of follow-up, and pathological type of participants. The outcomes extracted for network meta-analysis were complete ablation of BE, neoplastic progression of BE, and complete eradication of dysplasia.

## Risk of bias of individual studies

To assess the risk of bias (ROB), the two authors used the Cochrane Handbook version 5.1.0 tool [8] for RCTs. They evaluated seven domains, including randomized sequence generation, treatment allocation concealment, blinding of participants and personnel, incomplete outcome data, selective reporting, and other sources of bias. Studies with four or more high-risk domains were considered to have an overall high risk of bias. Meta-regression was conducted to examine the influence of risk of bias.

**Table 1. Overview of treatment groups.**

| | |
|---|---|
| **APC** | Argon plasma coagulation |
| **PDT** | Photodynamic therapy |
| **RFA** | Radiofrequency ablation |
| **SHAM/PBO** | Sham procedure/ Placebo |
| **Surveillance** | Only endoscopic surveillance |
| **PPI/H2RA** | Use only proton pump inhibitors or H2 receptor antagonists for treatment |
| **MPEC** | Multipolar electrocoagulation |
| **SURGERY** | Anti-reflux surgery |
| **NSAIDs** | Nonsteroidal anti-inflammatory drugs |
| **SRER** | Stepwise radical endoscopic resection |

## Statistical analysis

For each pairwise comparison and outcome, RRs with 95% CI were obtained as a measure of the association between interventions. Conventional meta-analyses were conducted using fixed- and random-effects models, with inverse variance for each outcome and comparison. The standard chi$^2$ test was used at a significance level of 0.1. Heterogeneity was considered significant when I$^2$ value exceeded 50%. Inconsistency was also evaluated since it is crucial in NWMs [9]. We constructed comparison-adjusted funnel plots and checked for symmetry to evaluate whether studies with smaller sample sizes among those included in the article influenced efficacy results. Effect modifiers were evaluated using meta-regression. Surface under cumulative ranking (SUCRA) values were used in rankograms for the intervention network, examining the cumulative rank probability for each intervention's efficacy compared to an ideal intervention that showed the best efficacy without doubt. SUCRA = 1 or 100% indicated the best efficacy achieved as a percentage [5,6,10]. Data were processed using Bayesian NMA software, i.e., Stata 16.0 (StataCorp, College Station, TX, USA). A p-value < 0.05 was significant for all tests, except for heterogeneity where the respective value was 0.1.

## Results

### Literature search and quality assessment

The literature search identified 1291 studies, from which 307 duplicates were removed. After screening and full-text review, 23 studies with a total of 1675 participants were included (Table 2) [11–33]. A flowchart illustrating the study selection is provided in Fig 1. The result of the quality assessment is shown in Fig 2, with six [16–19,21,26] out of 23 studies considered to have a high overall risk of bias, mainly due to the lack of allocation concealment or blinding to the treatment arms. As the meta-regression showed that excluding high-risk studies did not have a significant impact on the final results, we included all studies in the final analysis.

### Pair-wise meta-analysis

Pairwise meta-analysis and forest plots were conducted for comparisons with two or more studies, and forest plots are presented in S1 Fig. For complete ablation of BE, there was a significant difference in APC versus Surveillance risk ratio (RR) of 4.39; 95% CI 1.85, 10.41, favoring APC, and APC versus PPI/H2RA RR of 14.00; 95% CI 2.05, 95.56 favoring APC; PDT versus SHAM/PBO RR of 14.78; 95% CI 2.16, 101.16 favoring PDT, and PDT versus PPI/H2RA RR of 7.30; 95% CI 3.09, 17.25 favoring PDT. In terms of neoplastic progression, PPI/H2RA versus SURGERY RR was 9.12; 95% CI 1.19, 69.98 favoring SURGERY, RFA versus SHAM/PBO RR was 0.22; 95% CI 0.06, 0.81 favoring RFA, APC versus Surveillance RR was 0.18; 95% CI 0.05, 0.71 favoring APC, and PDT versus PPI/H2RA RR was 0.46; 95% CI 0.26, 0.81 favoring PDT. For complete eradication of dysplasia, there was a significant difference in RFA versus Surveillance (RR 3.52; 95% CI 2.40, 5.17), PDT versus PPI/H2RA (RR 4.11; 95% CI 2.28, 7.42), and RFA versus SHAM/PBO (RR 4.10; 95% CI 2.28, 7.37). S2 Fig is a funnel plot made for all three outcomes, which shows an acceptable distribution without any signs of publication bias.

### NMA results and meta-regression

The network map of therapeutic interventions for BE included in 23 RCTs is illustrated in Figs 3A1, 3A2, 4A and 5A. The node size reflects the number of patients allocated to each treatment, while edge thickness is proportional to precision, i.e., the inverse of variance for each direct comparison. Four direct comparisons were possible for complete ablation of non-

**Table 2. Basic demographic data of RCT studies included in the review.**

| Study ID | Country | Sample size | Gender ratio(male,%) | Mean age(years) | Follow-up, months | Pathological type | Treatment regimens |
|---|---|---|---|---|---|---|---|
| Panjehpour M 2000 [12] | USA | 60 | 83.3% | \ | 10 | LGD+HGD+EC | PDT |
| Ackroyd R 2000 [11] | UK | 36 | 83.3% | 54.8±9.7 | 24 | LGD | PDT, PBO |
| Kahaleh M 2002 [13] | Belgium | 39 | 76.9% | 63.7±8.7 | 1 | NDBE+LGD | APC |
| Parrilla P 2003 [14] | Spain | 101 | 71.3% | 45.3±14.4 | 72 | NDBE+LGD | PPI/H2RA, SURGERY |
| Ackroyd R 2004 [15] | UK | 40 | 80.0% | 48.9±5.1 | 12 | NDBE+LGD | APC, Surveillance |
| Hage M 2004 [16] | Netherlands | 40 | 77.5% | 59.6±6.3 | 12 | NDBE+LGD | PDT, APC |
| Kelty CJ 2004 [17] | UK | 72 | 80.1% | 59.0±11.9 | 12 | NDBE | PDT, APC |
| Overholt BF 2005 [18] | USA | 208 | 84.6% | 66.5±10.8 | 6 | HGD | PDT, PPI |
| Ragunath K 2005 [19] | UK | 26 | 80.8% | 61.2±13.2 | 12 | LGD+HGD | APC, PDT |
| Sharma P 2006 [20] | USA | 35 | 97.1% | 61.1±12.4 | 24 | NDBE+LGD | APC, MPEC |
| Bright T 2007 [21] | Australia | 40 | 80.0% | 58.0±9.6 | 60 | NDBE+LGD | APC, Surveillance |
| Heath EI 2007 [22] | USA | 100 | 87.0% | 67±7.2 | 12 | LGD+HGD | NSAIDs, PBO |
| Bright T 2009 [23] | Australia | 54 | 70.4% | 59.1±7.6 | 12 | NDBE+LGD | APC, Surveillance |
| Shaheen NJ 2009 [24] | USA | 127 | 86.6% | 66.0±1.7 | 12 | LGD+HGD | RFA, SHAM |
| L Zhang 2009 [33] | China | 35 | 62.9% | 53.3±10.7 | 12 | NDBE+LGD | APC, PPI |
| van Vilsteren FG 2011 [25] | Netherlands | 47 | 85.1% | 67.6±8.5 | 24 | HGD+EC | SRER, RFA |
| Sie C 2013 [26] | Australia | 129 | \ | 61.5±9.3 | 12 | NDBE+LGD | APC, Surveillance |
| Manner H 2014 [27] | Germany | 63 | 90.5% | 63.0±9.9 | 36 | HGD+EC | APC, Surveillance |
| Phoa KN 2014 [28] | Europe | 136 | 85.3% | 63±9.5 | 36 | LGD | RFA, Surveillance |
| Kohoutova D 2018 [29] | UK | 58 | \ | 68.1±9.0 | 60 | HGD+EC | PDT |
| Peerally MF 2019 [30] | UK | 76 | 84.2% | 70.0±9.2 | 12 | HGD+EC | RFA, APC |
| Wronska E 2021 [32] | Poland | 71 | 76.1% | 61.1±4.2 | 1.5 | LGD | APC |
| Barret M 2021 [31] | France | 82 | 92.7% | 62.3±10.0 | 36 | LGD | RFA, Surveillance |

NDBE: Non-dysplastic BE. LGD: Low grade dysplasia. HGD: High grade dysplasia. EC: Esophageal cancer. APC: Argon plasma coagulation. PDT: Photodynamic therapy. RFA: Radiofrequency ablation. SHAM/PBO: Sham procedure/ Placebo. Surveillance: Endoscopic surveillance. PPI/H2RA: Use only proton pump inhibitors or H2 receptor antagonists for treatment. MPEC: Multipolar electrocoagulation. SURGERY: Anti-reflux surgery. NSAIDs: Nonsteroidal anti-inflammatory drugs. SRER: Stepwise radical endoscopic resection.

dysplastic BE(NDBE), six for complete ablation of BE with low-grade dysplasia (LGD) or high-grade dysplasia (HGD) or esophageal cancer (EC), 11 for neoplastic progression, and six for complete eradication of dysplasia. Meta-regression was performed using the aforementioned effect modifiers as a covariate, and the deviance information criterion (DIC) was compared to the model. The difference was less than one and therefore insignificant, so no covariates were used in the analysis.

## Complete ablation of BE(NDBE)

The forest plot with APC as the comparator is presented in Fig 3B1. Surveillance and PPI/H2RA are significantly inferior to APC, regarding complete ablation of NDBE. PDT and MPEC are also performing worse, but without statistical significance. The SUCRA plot (Fig 3C1) confirms these results, with the lowest ranked treatments being PPI/H2RA (SUCRA value 8.5%) and Surveillance (SUCRA value 31.4%). The highest ranked treatments are APC (SUCRA value 84.3%) and MPEC (SUCRA value 67.5%). The league table of comparative efficacies in Fig 3D1 shows the same trend, demonstrating that APC performs significantly better than PPI/H2RA and Surveillance.

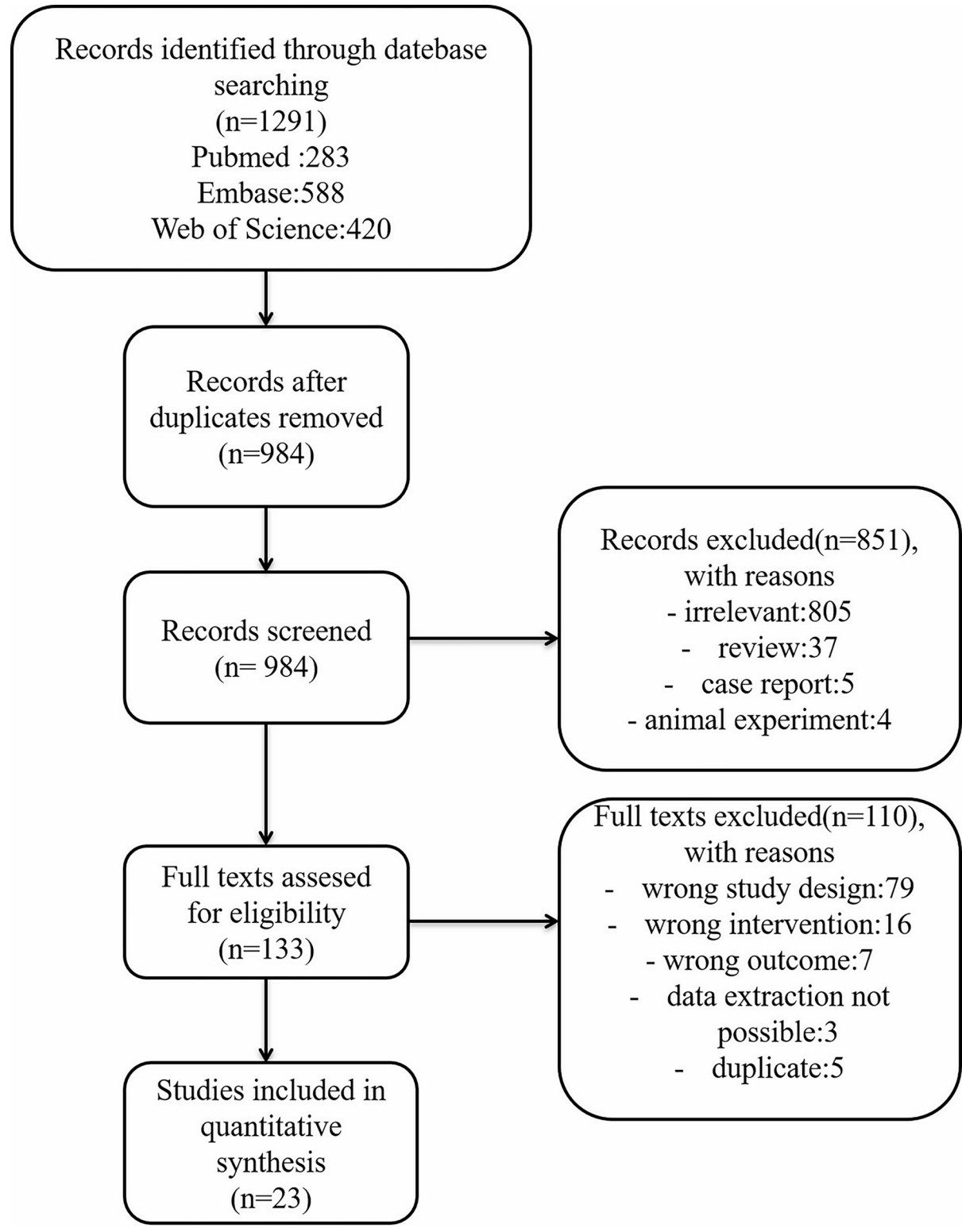

**Fig 1. PRISMA flowchart for searching and selecting eligible studies.**

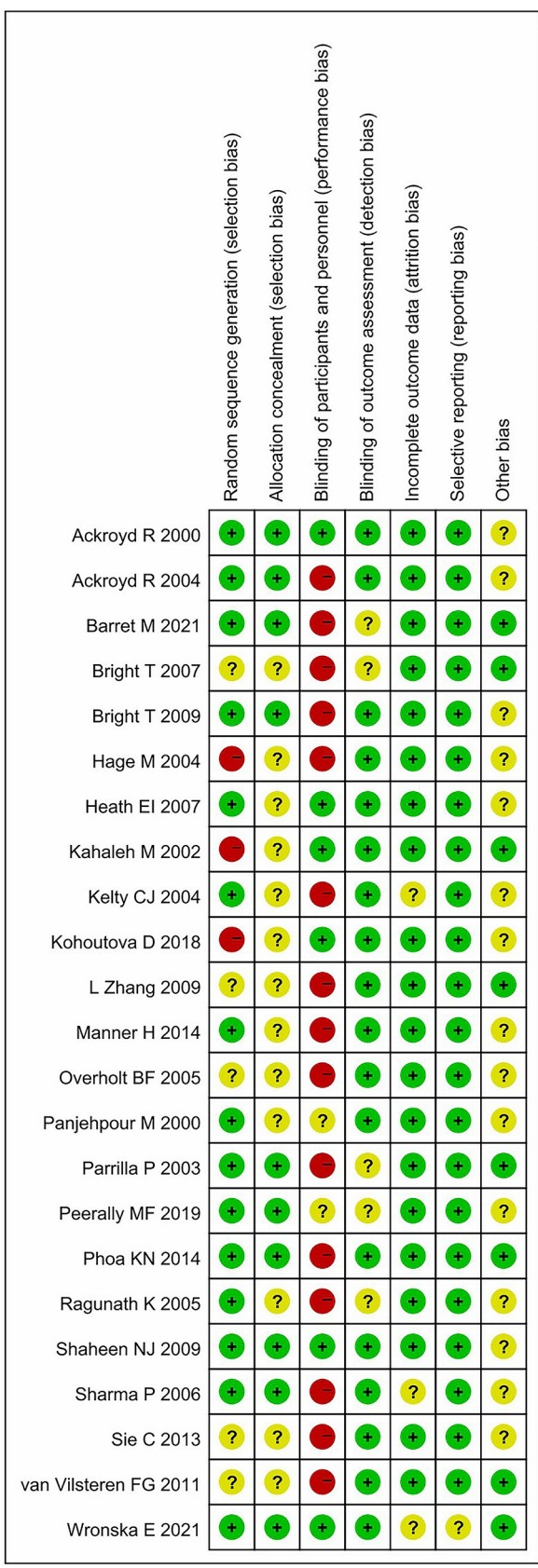

**Fig 2. Risk of bias summary.** Six out of 23 studies considered to have a high overall risk of bias.

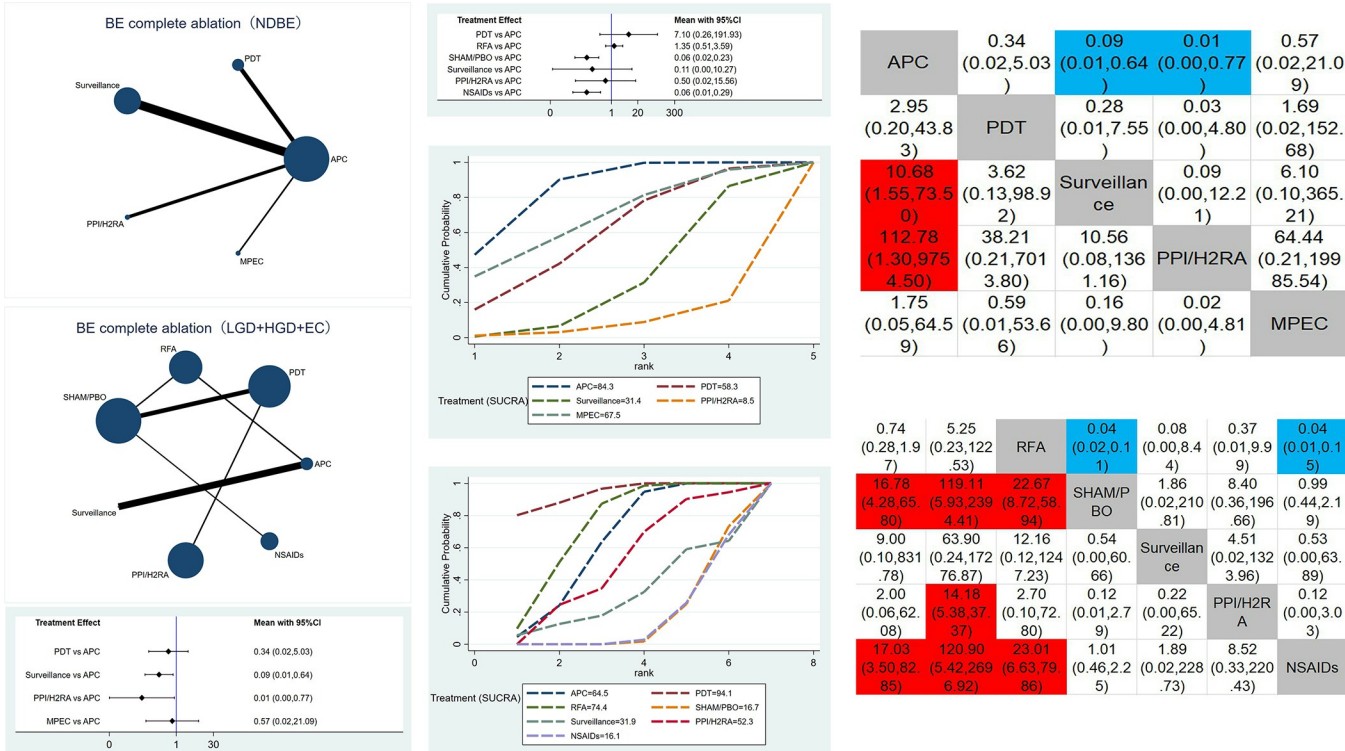

**Fig 3. A1. Geometry of the network for complete ablation of BE(NDBE).** Each dot represents a treatment modality. Lines between dots shows a direct comparison, for example, a study comparing these two treatments. Thicker lines indicate more studies. **A2. Geometry of the network for complete ablation of BE(LGD+HGD+EC).** Each dot represents a treatment modality. Lines between dots shows a direct comparison, for example, a study comparing these two treatments. Thicker lines indicate more studies. **B1. Forest plot for complete ablation of BE(NDBE).** Argon plasma coagulation compared against the other treatments. Results are presented as difference in complete ablation of BE with 95% confidence intervals. **B2. Forest plot for complete ablation of BE(LGD+HGD+EC).** Argon plasma coagulation compared against the other treatments. Results are presented as difference in complete ablation of BE with 95% confidence intervals. **C1. SUCRA plot for complete ablation of BE(NDBE).** SUCRA, surface under the curve cumulative ranking probabilities, shows probability of ranking for each treatment illustrated by graphs. **C2. SUCRA plot for complete ablation of BE(LGD+HGD+EC).** SUCRA, surface under the curve cumulative ranking probabilities, shows probability of ranking for each treatment illustrated by graphs. **D1. Comparative efficacy ranking league matrix (Complete ablation of BE(NDBE)).** Comparison of all treatment in the network, combined indirect and direct if, available. Results are presented as difference in complete ablation of BE with 95% confidence intervals. Red color shows an advantage for treatment, blue color disadvantage. **D2. Comparative efficacy ranking league matrix (Complete ablation of BE(LGD+HGD+EC)).** Comparison of all treatment in the network, combined indirect and direct if, available. Results are presented as difference in complete ablation of BE with 95% confidence intervals. Red color shows an advantage for treatment, blue color disadvantage.

## Complete ablation of BE(LGD+HGD+EC)

For BE with low-grade dysplasia or high-grade dysplasia or esophageal cancer, NSAIDs and SHAM/PBO are significantly inferior to APC as shown in the forest plot, Fig 3B2. The SUCRA plot (Fig 3C2) reveals that PDT (SUCRA value 94.1%), RFA (SUCRA value 74.4%), and APC (SUCRA value 64.5%) are the highest ranked treatments, whereas NSAIDs (SUCRA value 16.1%) and SHAM/PBO (SUCRA value 16.7%) are the lowest ranked treatments. The league table of comparative efficacies in Fig 3D2 shows the same trend, demonstrating that NSAIDs and SHAM/PBO are significantly inferior to PDT, RFA, and APC. Additionally, PPI/H2RA is clinically different when compared to PDT.

## Neoplastic progression

The forest plot with APC as a comparator is presented in Fig 4B. While SURGERY, RFA, and MPEC perform better, the results are not statistically significant. These findings are confirmed

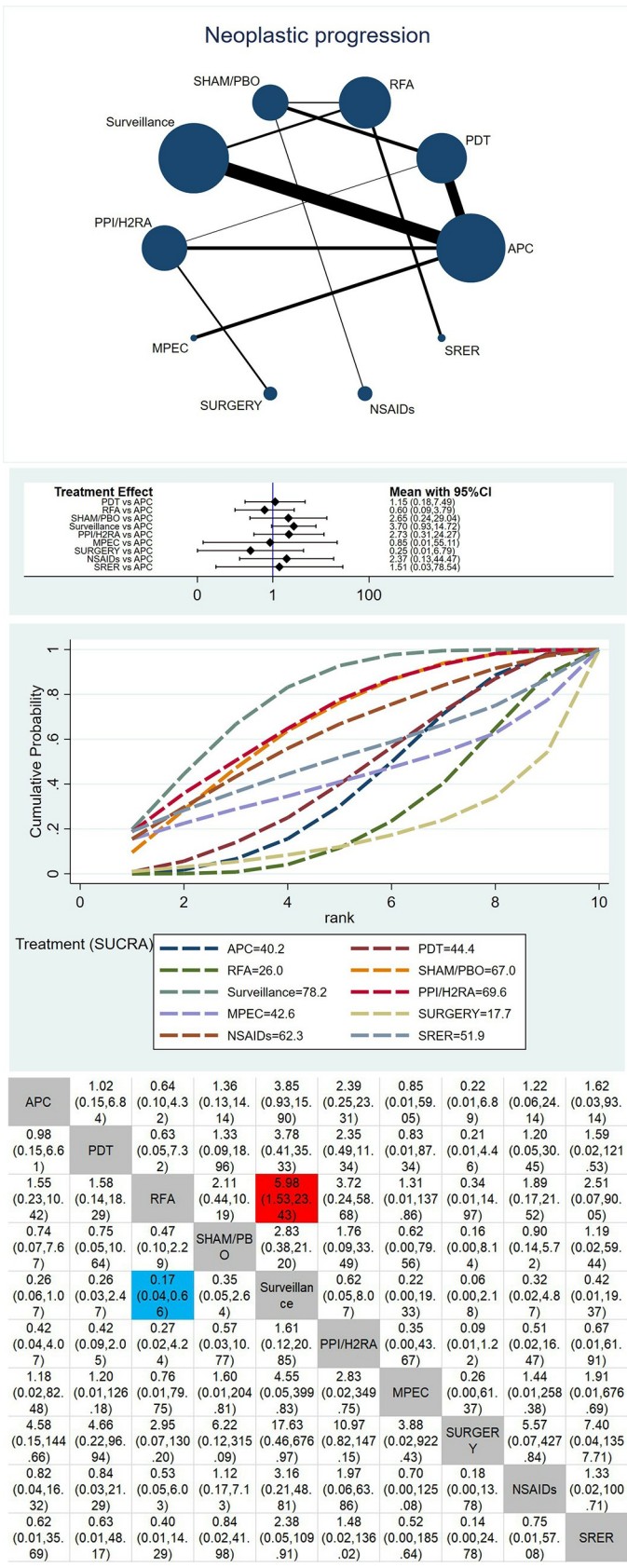

**Fig 4. A. Geometry of the network for neoplastic progression.** Each dot represents a treatment modality. Lines between dots shows a direct comparison, for example, a study comparing these two treatments. Thicker lines indicate more studies. **B. Forest plot for neoplastic progression.** Argon plasma coagulation compared against the other treatments. Results are presented as difference in neoplastic progression with 95% confidence intervals. **C. SUCRA plot for neoplastic progression.** SUCRA, surface under the curve cumulative ranking probabilities, shows probability of ranking for each treatment illustrated by graphs **D. Comparative efficacy ranking league matrix (Neoplastic progression).** Comparison of all treatment in the network, combined indirect and direct if, available. Results are presented as difference in neoplastic progression with 95% confidence intervals. Red color shows an advantage for treatment, blue color disadvantage.

in the SUCRA plot (Fig 4C), which shows that the lowest ranked treatments are SURGERY (SUCRA value 17.7%), RFA (SUCRA value 26.0%), and APC (SUCRA value 40.2%). The highest ranked treatments are Surveillance (SUCRA value 78.2%), PPI/H2RA (SUCRA value 69.6%), and SHAM/PBO (SUCRA value 67.0%). The league table of comparative efficacies in Fig 4D demonstrates that RFA performs significantly better than Surveillance.

## Complete eradication of dysplasia

Surveillance and SHAM/PBO are significantly inferior to APC as shown in the forest plot, Fig 5B. Moreover, PDT is performing better, although without statistical significance. The SUCRA plot (Fig 5C) indicates that the highest ranked treatments are PDT (SUCRA value 91.2%), APC (SUCRA value 69.8%), and SRER (SUCRA value 65.5%). The lowest ranked treatments are Surveillance (SUCRA value 1.0%) and SHAM/PBO (SUCRA value 17.3%). The league table of comparative efficacies in Fig 5D shows the same trend. Firstly, it demonstrates that Surveillance is significantly inferior to all other treatments except for SHAM/PBO. Secondly, PDT, APC, and RFA are significantly better than SHAM/PBO. Lastly, PDT is significantly better than PPI/H2RA.

## Discussion

The treatment of BE remains a controversial topic, particularly for BE with different degrees of metaplasia. In recent years, the focus of BE management has shifted towards complete ablation of BE, prevention of neoplastic progression, and complete eradication of dysplasia. Currently, there is an urgent need for high-quality research comparing different treatment options for BE. Network meta-analysis is a novel analytical method that can compare multiple interventions simultaneously, providing a larger statistical basis. In this manuscript, we have included almost all commonly used treatment methods and examined their effectiveness in complete ablation of BE, prevention of neoplastic progression, and complete eradication of dysplasia. To our knowledge, this is the first Bayesian network meta-analysis comparing all treatment options for curing BE.

Our findings indicate that APC (SUCRA 84.3%) is the best regimen and performs significantly better than Surveillance and PPI/H2RA in terms of complete ablation of non-dysplastic BE, which is clinically significant. Recently, a novel hybrid ablation method known as hybrid APC has emerged, building upon APC. It has been reported that hybrid APC boasts a lower complication rate compared to traditional APC [34]. Furthermore, its safety and effectiveness have been further affirmed in recent prospective studies [35,36]. MPEC (SUCRA 67.5%) has also shown excellent efficacy, which is a technique used in endoscopy mainly for the treatment of gastrointestinal bleeding. Studies have shown that it can be an effective method for treating dysplasia and early-stage cancer in patients with Barrett's esophagus, offering a less invasive alternative to surgery [37]. This procedure involves using a special device with multiple electrodes to deliver controlled and targeted heat to the abnormal esophageal tissue, with the goal

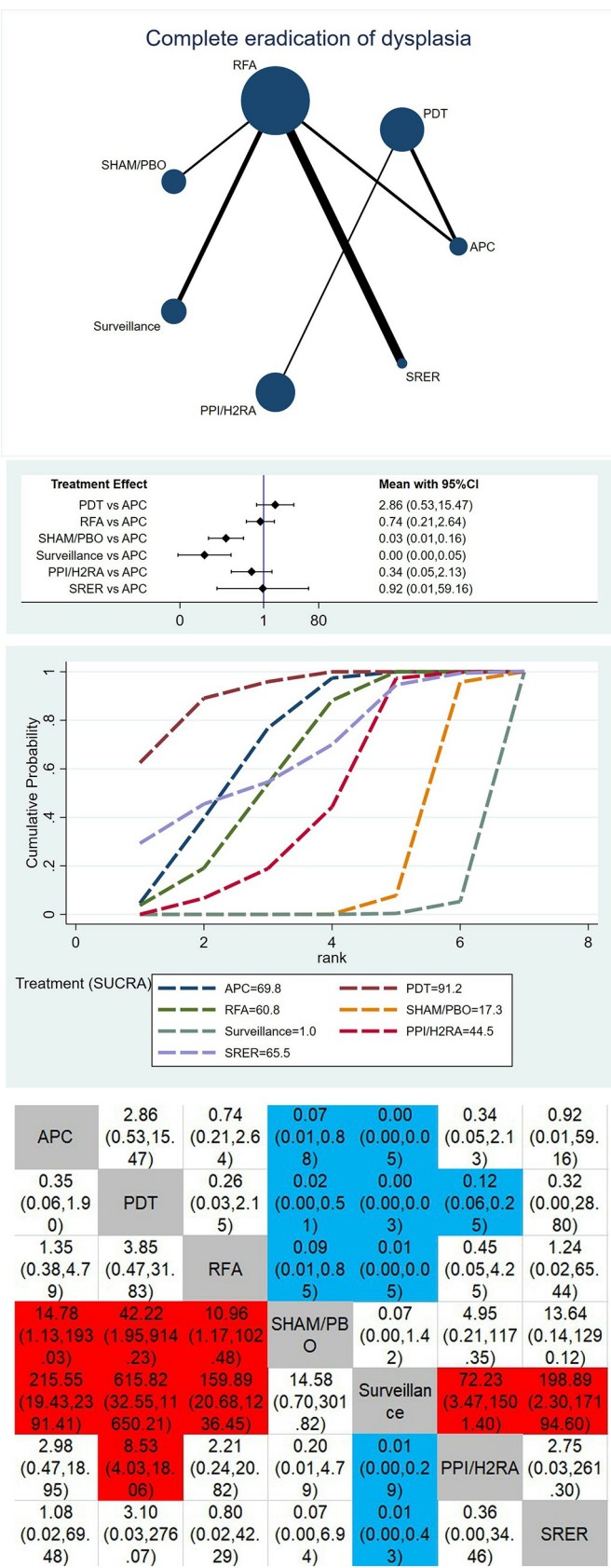

**Fig 5. A. Geometry of the network for complete eradication of dysplasia.** Each dot represents a treatment modality. Lines between dots shows a direct comparison, for example, a study comparing these two treatments. Thicker lines indicate more studies. **B. Forest plot for complete eradication of dysplasia.** Argon plasma coagulation compared against the other treatments. Results are presented as difference in complete eradication of dysplasia with 95% confidence intervals. **C. SUCRA plot for complete eradication of dysplasia.** SUCRA, surface under the curve cumulative ranking probabilities, shows probability of ranking for each treatment illustrated by graphs. **D. Comparative efficacy ranking league matrix (Complete eradication of dysplasia).** Comparison of all treatment in the network, combined indirect and direct if, available. Results are presented as difference in complete eradication of dysplasia with 95% confidence intervals. Red color shows an advantage for treatment, blue color disadvantage.

of destroying the abnormal cells and allowing healthy cells to regenerate. Several European and American guidelines recommend long-term endoscopic surveillance for non-dysplastic BE patients every 3–5 years [38,39]. It is believed that the progression from Barrett's esophagus to adenocarcinoma occurs through low-grade and high-grade dysplasia. It has been suggested that non-dysplastic BE patients have a low risk of developing carcinoma, and thus it is inappropriate to treat them [11,40]. Nevertheless, SHARMA P and her colleagues have found that a subset of patients with intestinal metaplasia may rapidly advance to invasive carcinoma without exhibiting the traditionally observed progression through dysplasia [41]. Therefore, an intervention that may reduce this risk should be considered for all patients, consistent with our NMA results. Additionally, although controlling esophageal pH by PPIs may be associated with regression of BE, control of pH alone may not be sufficient to cause significant regression [42,43].

Regarding complete ablation of BE with low-grade dysplasia or high-grade dysplasia or esophageal cancer, we found that PDT is the best regimen (SUCRA value 94.1%), with a significantly higher SUCRA value compared to RFA (74.4%), APC (64.5%), PPI/H2RA (52.3%), Surveillance (31.1%), SHAM/PBO (16.7%), and NSAIDs (16.1%). According to European guidelines [44], endoscopic intervention is recommended for BE with low-grade dysplasia, high-grade dysplasia, or esophageal cancer, which aligns with our findings. The notable distinction, however, lies in the exceptional efficacy demonstrated by PDT in our study. In addition, in the analysis of complete eradication of dysplasia, SUCRA plots also showed a trend in ranking PDT highest (SUCRA value 91.2%). PDT is often combined with other treatments, such as endoscopic mucosal resection. In a five-year multicenter study, Overholt et al. confirmed the efficacy and safety of PDT for BE with HGD, which can produce long-term ablation of HGD and reduce the potential impact of cancer [45]. According to several studies [46–48], PDT has an ablation rate between 60% to 80%. Furthermore, multiple outcome studies have shown that PDT is more cost-effective than surgery and surveillance [49–51].

For neoplastic progression, RFA (SUCRA value 26.0%) and SURGERY (SUCRA value 17.7%) perform significantly better than Surveillance. Recent guidelines recommend RFA for the eradication of remaining Barrett's epithelium after endoscopic resection of visible abnormalities containing any degree of dysplasia or neoplasia [39]. A recent meta-analysis by Wang Y et al. revealed that, compared with endoscopic surveillance, RFA decreased the risk of BE-LGD progression to BE-HGD or EAC by up to 75% [52]. Fundoplication aims to reduce gastroesophageal reflux disease, which is the greatest modifiable risk factor in the metaplasia-EAC sequence [53]. Compared with anti-reflux medication, a recent study suggests that fundoplication may be superior in preventing histopathological progression of BE to EAC [54].

The Cochrane Handbook version 5.1.0 tool for assessing ROB in RCTs has been followed in making this manuscript to ensure high quality. All researchers' evaluations of this study gave a high confidence result. However, caution must still be taken when interpreting the results as six studies had a high overall risk of bias. Although the sensitivity analysis did not indicate any issues, it might give an imprecise results since some outcomes were only

represented by a few studies. Another limitation of this study cannot be ignored is the heterogeneity between the studies. While heterogeneity in the participant population itself is considered low, differences in the diagnostic criteria of BE in Japan and Western countries were not included in this NMA due to the lack of RCTs from Japan.

Overall, endoscopic surveillance alone or PPI/H2RA alone may not be sufficient in managing BE, even in non-dysplastic BE. APC had excellent efficacy in treating non-dysplastic BE. For BE with low-grade dysplasia or high-grade dysplasia or esophageal cancer, PDT may be the optimal intervention as it incites regression of BE metaplasia and prevents future progression of BE to dysplasia and EC.

## Supporting information

**S1 Checklist. PRISMA 2020 checklist.**
(DOCX)

**S1 Fig. Pairwise meta-analysis and forest plots.**
(TIF)

**S2 Fig. Funnel plot.**
(TIF)

**S1 Table. The data of meta-analysis.**
(XLSX)

## Author Contributions

**Conceptualization:** Wensheng Pan.

**Data curation:** Qinlin Zhang.

**Formal analysis:** Qinlin Zhang, Xin Jin.

**Funding acquisition:** Qinlin Zhang.

**Investigation:** Qinlin Zhang, Xin Jin, Yize Ying.

**Methodology:** Qinlin Zhang, Xin Jin, Wensheng Pan.

**Project administration:** Jiyong Jing.

**Resources:** Wensheng Pan.

**Software:** Miya Li, Xin Jin, Ruhong Zhou.

**Supervision:** Xueping Wu, Jiyong Jing, Wensheng Pan.

**Validation:** Miya Li, Xin Jin, Yize Ying, Xueping Wu.

**Visualization:** Ruhong Zhou.

**Writing – original draft:** Qinlin Zhang.

**Writing – review & editing:** Qinlin Zhang.

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
