## [Decision Letter · Decision Letter 0]

26 Dec 2023

PONE-D-23-16418Comparison of interventions for Barrett's esophagus: a network meta-analysisPLOS ONE

Dear Dr. Pan,

Thank you for submitting your manuscript to PLOS ONE. After careful consideration, we feel that it has merit but does not fully meet PLOS ONE’s publication criteria as it currently stands. Therefore, we invite you to submit a revised version of the manuscript that addresses the points raised during the review process.

We look forward to receiving your revised manuscript.

Kind regards,

Alejandro Piscoya

Academic Editor

PLOS ONE

Journal Requirements:

https://www.dovepress.com/front_end/getfile.php?fileID=26200

https://onlinelibrary.wiley.com/doi/10.1111/j.1365-2036.2004.02277.x

In your revision ensure you cite all your sources (including your own works), and quote or rephrase any duplicated text outside the methods section. Further consideration is dependent on these concerns being addressed.

3. Please include your tables as part of your main manuscript and remove the individual files. Please note that supplementary tables (should remain/ be uploaded) as separate "supporting information" files.

“This is an unfunded study”

Reviewers' comments:

Reviewer's Responses to Questions

**Comments to the Author**

1. Is the manuscript technically sound, and do the data support the conclusions?

Reviewer #1: Partly

Reviewer #2: Yes

Reviewer #3: Partly

Reviewer #4: Partly

2. Has the statistical analysis been performed appropriately and rigorously? 

Reviewer #1: N/A

Reviewer #2: Yes

Reviewer #3: Yes

Reviewer #4: Yes

3. Have the authors made all data underlying the findings in their manuscript fully available?

Reviewer #1: No

Reviewer #2: Yes

Reviewer #3: Yes

Reviewer #4: Yes

4. Is the manuscript presented in an intelligible fashion and written in standard English?

Reviewer #1: Yes

Reviewer #2: Yes

Reviewer #3: Yes

Reviewer #4: Yes

5. Review Comments to the Author

Reviewer #1: 1. Non-dysplastic BE does not need eradication treatment based on cost-effects ratio. The authors suggest a different opinion refering to referance 37, however, the article can not be searched. A article written by Prateek Sharma,et al. <dysplasia a="" and="" barrett="" cancer="" cohort="" in="" large="" multicenter="" of="" patients="" with="">, seems the article the authors referred, according to the observation from Prateek Sharma, there was only 18 patients of 1376 objectives had at least 2 initial consecutive endoscopies with biopsies revealing non-dysplastc mucosa. And the inaccuracy operation of endoscopic biopsy making an imoact on the result should be considered, then, this is not strong evident supporting authors opinion. It is better for the authors to review more publishes correctly.

2. Endoscopic surveillace is a effective method to detect early carcinoma also as a uesful management for patients who accepted endoscopic eradication therapy or other treatment. But, It should not be compared with other complete eradication treatment.

3. For now, The best treatment for BE should based on the grade of dysplasia and the widths and depths of the lesion. For example, Endoscopic mucosal resection (EMR) is a widely used technique for diagnosing and

eradicating superficial BE dysplasia and neoplasia, Radiofrequency ablation (RFA) can be employed to eradicate circumferential areas of dysplastic BE.

4.Photodynamic Therapy is usually applied to esophageal cancer treatment. For BE, it is a option, but not commonly accepted.

5. The authors research seems a little contributions for BE treatment reseach. Obsolete publishes included might be a reason.</dysplasia>

Reviewer #2: The author analyzed the advantages and disadvantages of different treatment methods for Barrett's esophagus using a net META method. The literature included in the study was all RCT studies, with small differences in sample size and high reliability in each study. Preliminary analysis has been conducted on the advantages of current methods for treating Barrett's esophagus. However, there are still some shortcomings in this article. In the discussion section, the current Barrett's esophageal treatment guidelines were not combined to point out the similarities and differences between your research and the guidelines.

Reviewer #3: This meta-analysis deals with a very important topic, the treatment of Barrett's esophagus. The authors compare existing treatment options in order to evaluate them on the basis of available RCTs.

The present manuscript still needs some fine-tuning in a few points:

Minor remarks:

1. Please insert a space before the brackets when introducing an abbreviation.

2. In the methods section "study selection", please define what you consider as a small sample size.

3. Statistical analysis: Here you write that it was investigated to what extent a small-sized trial influenced the results. However, in the study selection section small sized trials were excluded, please explain the discrepancy between these two points.

4. Table 1: It would be nice, if you could explain in the legend, what is meant by wrong outcome, as this term is slightly misleading.

5. MPEC is not introduced in the text.

6. For figure 2: Please consider indicating, which of the 23 studies you have considered as potentially biased.

7. It is slightly irritating, that Fig. 4A and 5A are mentioned way earlier in the text then the following numbers.

8. Figure legend 2 needs a little more information what is to see in the figure.

9. Reference in text to supplement figure 2 is missing.

10. Also discuss newer developments like Hybrid-APC:

Shimizu, T., Samarasena, J. B., Fortinsky, K. J., Hashimoto, R., Chehade, N. E. H., Chin, M. A., ... & Chang, K. J. (2021). Benefit, tolerance, and safety of hybrid argon plasma coagulation for treatment of Barrett's esophagus: US pilot study. Endoscopy International Open, 9(12), E1870-E1876.

Knabe, M., Beyna, T., Rösch, T., Bergman, J., Manner, H., May, A., ... & Ehlken, H. (2022). Hybrid APC in Combination With Resection for the Endoscopic Treatment of Neoplastic Barrett's Esophagus: A Prospective, Multicenter Study. The American journal of gastroenterology, 117(1), 110-119.

11. In your overall conclusion I am missing a statement regarding surgery as a therapeutic option.

Reviewer #4: This appears to be an article that could serve as an excellent guideline for the treatment of Barrett's esophagus (BE). However, there are a few points of curiosity that I would like to address:

1. In the "study selection", the article divided the treatments into 10 major groups. I wonder if it could have been possible to group similar treatments together to create fewer groups.

2. In the "study selection", it was mentioned that studies with small sample sizes were excluded. It would be helpful to specify the criteria used to determine what is considered a small sample size.

3. While there seems to be a fundamental difference in the meaning of "complete ablation of BE" and "complete eradication of dysplasia," in practical terms, they appear to have very similar criteria. It would be beneficial to mention the rationale for separating these two, as they seem quite similar in clinical application.

4. In Figure 1, the number of cases classified as "irrelevant" is quite high (805 cases). It would be helpful if the specific criteria used to make this determination were explicitly stated.

5. Generally, dysplasia and esophageal adenocarcinoma are assessed using different criteria. If the article refers to "complete ablation of BE" and places adenocarcinoma on the same level as dysplasia, it would be useful to explain the reasoning behind this.

6. If the article suggests that adenocarcinoma can progress directly from metaplasia without going through dysplasia, it might be more helpful to mention this in the introduction or study design section rather than just in the discussion. This could help reduce confusion.

7. If the above-mentioned points of concern are addressed, this article has the potential to be clinically very helpful.

6. PLOS authors have the option to publish the peer review history of their article (what does this mean?). If published, this will include your full peer review and any attached files.

Reviewer #1: No

Reviewer #2: No

Reviewer #3: No

Reviewer #4: No

---

## [Author Response · Author response to Decision Letter 0]

3 Feb 2024

Dear Editors and Reviewers:

Thank you for your letter and for the reviewers’ comments concerning our manuscript entitled “Comparison of interventions for Barrett's esophagus: a network meta-analysis” (ID: PONE-D-23-16418). Those comments are all valuable and very helpful for revising and improving our paper, as well as the important guiding significance to our researches. We have studied comments carefully and have made correction which we hope meet with approval.

SUGGESTIONS FROM EDITOR（1. Please ensure that your manuscript meets PLOS ONE's style requirements, including those for file naming. The PLOS ONE style templates can be found at

https://www.dovepress.com/front_end/getfile.php?fileID=26200

https://onlinelibrary.wiley.com/doi/10.1111/j.1365-2036.2004.02277.x

In your revision ensure you cite all your sources (including your own works), and quote or rephrase any duplicated text outside the methods section. Further consideration is dependent on these concerns being addressed.

3. Please include your tables as part of your main manuscript and remove the individual files. Please note that supplementary tables (should remain/ be uploaded) as separate "supporting information" files.

“This is an unfunded study”

5. Please include captions for your Supporting Information files at the end of your manuscript, and update any in-text citations to match accordingly. Please see our Supporting Information guidelines for more information: http://journals.plos.org/plosone/s/supporting-information. ）

Our response to the general comments from the editor：

1. The manuscript has been revised in accordance with PLOS ONE's style requirements.

2. The minor issue of overlapping text with the previously mentioned publications has been resolved.

3. We have integrated the tables into the manuscript and removed the individual files.

4. The authors did not receive any specific funding for this work. We have included our amended statements within cover letter.

5. We have added captions for the Supporting Information files at the end of our manuscript and adjusted any in-text citations accordingly.

Revised portion are marked in red in the paper. The main corrections in the paper and the responds to the reviewer’s comments are as flowing:

Reviewer #1:

1. Non-dysplastic BE does not need eradication treatment based on cost-effects ratio. The authors suggest a different opinion refering to referance 37, however, the article can not be searched. A article written by Prateek Sharma,et al. , seems the article the authors referred, according to the observation from Prateek Sharma, there was only 18 patients of 1376 objectives had at least 2 initial consecutive endoscopies with biopsies revealing non-dysplastc mucosa. And the inaccuracy operation of endoscopic biopsy making an impact on the result should be considered, then, this is not strong evident supporting authors opinion. It is better for the authors to review more publishes correctly.

The author’s answer: We are truly grateful for your professional review of our article. As per your request, Reference 37 can be accessed at the following address: https://www.researchgate.net/publication/246664693_Progression_of_Barrett's_esophagus_to_high_grade_dysplasia_and_cancer_preliminary_results_of_the_BEST_Barrett's_Esophagus_Study_Trial Although the potential impact of inaccuracies in endoscopic biopsy operations on the results should be considered, it is undeniable that this extensive study suggests a need for caution regarding the rapid progression from intestinal metaplasia to invasive carcinoma.

2. Endoscopic surveillace is a effective method to detect early carcinoma also as a uesful management for patients who accepted endoscopic eradication therapy or other treatment. But, It should not be compared with other complete eradication treatment.

The author’s answer: Thank you once more for your encouraging feedback and insightful suggestions on enhancing the quality of our manuscript. Regarding patients with non-dysplastic Barrett's esophagus, options such as endoscopic eradication or alternative treatments, along with regular endoscopic surveillance without additional intervention, are considered as potential treatment alternatives. Our aim in comparing these treatment options is to identify the most beneficial approach for the patients.

3. For now, The best treatment for BE should based on the grade of dysplasia and the widths and depths of the lesion. For example, Endoscopic mucosal resection (EMR) is a widely used technique for diagnosing and eradicating superficial BE dysplasia and neoplasia, Radiofrequency ablation (RFA) can be employed to eradicate circumferential areas of dysplastic BE.

The author’s answer: I wholeheartedly agree with your opinion. However, as evidenced by the studies presented here, diverse treatment approaches are being employed for the management of patients with BE, perhaps indicative of the absence of a clear consensus. We aim to elucidate the relative priorities of these treatment options through rigorous scientific statistical methods.

4. Photodynamic Therapy is usually applied to esophageal cancer treatment. For BE, it is a option, but not commonly accepted.

The author’s answer: Thank you for your feedback. We aim to use our research to identify and advocate for treatments that are more effective than others and promote their widespread use.

5. The authors research seems a little contributions for BE treatment reseach. Obsolete publishes included might be a reason.

The author’s answer: We appreciate your constructive feedback. We will make the required revisions to our manuscript to ensure the credibility of our findings.

Reviewer #2:

The author analyzed the advantages and disadvantages of different treatment methods for Barrett's esophagus using a net META method. The literature included in the study was all RCT studies, with small differences in sample size and high reliability in each study. Preliminary analysis has been conducted on the advantages of current methods for treating Barrett's esophagus. However, there are still some shortcomings in this article. In the discussion section, the current Barrett's esophageal treatment guidelines were not combined to point out the similarities and differences between your research and the guidelines.

The author’s answer: On behalf of all the contributing authors, I would like to express our sincere appreciation for your constructive comments regarding our article. Your comments have been immensely valuable and have greatly aided in enhancing the quality of our work. In response to your feedback, we have made substantial modifications to our manuscript to bolster the credibility of our results. In this revised version, all changes have been clearly highlighted in red within the document. We are truly grateful for your positive feedback and invaluable suggestions, which have undoubtedly contributed to the overall improvement of our manuscript.

Reviewer #3:

1. Please insert a space before the brackets when introducing an abbreviation.

The author’s answer: We would like to express our gratitude for the time you've invested and for granting us the opportunity to enhance the manuscript. We have addressed the formatting issues in the article.

2. In the methods section "study selection", please define what you consider as a small sample size.

The author’s answer: We define small sample size studies as those with less than 20.

3. Statistical analysis: Here you write that it was investigated to what extent a small-sized trial influenced the results. However, in the study selection section small sized trials were excluded, please explain the discrepancy between these two points.

The author’s answer: To ensure the quality of the included Randomized Controlled Trials (RCTs), we excluded studies with small sample sizes below 20. Additionally, we constructed comparison-adjusted funnel plots and checked for symmetry to evaluate whether studies with smaller sample sizes among those included in the article influenced efficacy results.

4. Table 1: It would be nice, if you could explain in the legend, what is meant by wrong outcome, as this term is slightly misleading.

The author’s answer: Due to the variety of techniques and treatment regimens employed for Barrett's esophagus, we categorized all treatments into 10 major groups based on the selected studies, as detailed in Table 1.

5. MPEC is not introduced in the text.

The author’s answer: Thank you for your suggestion. We have included an introduction to MPEC in the article and highlighted it in red.

6. For figure 2: Please consider indicating, which of the 23 studies you have considered as potentially biased.

The author’s answer: Based on your valuable suggestions, we have incorporated six studies with a high risk of bias into the quality assessment section of the article and highlighted them using red font. Studies with four or more high-risk domains were deemed to possess an overall high risk of bias.

7. It is slightly irritating, that Fig. 4A and 5A are mentioned way earlier in the text then the following numbers.

The author’s answer: Thank you for your comments, which are greatly appreciated. I'd like to clarify that in our effort to systematically compare the effectiveness of various treatment options, we have established four different outcome indicators. To enhance readability for our readers, we have standardized the serial numbers of the associated pictures for each outcome indicator.

8. Figure legend 2 needs a little more information what is to see in the figure.

The author’s answer: Thank you for your comment, which we have supplemented in the manuscript.

9. Reference in text to supplement figure 2 is missing.

The author’s answer: We sincerely thank you for your valuable feedback that we have used to improve the quality of our manuscript.

10. Also discuss newer developments like Hybrid-APC:

Shimizu, T., Samarasena, J. B., Fortinsky, K. J., Hashimoto, R., Chehade, N. E. H., Chin, M. A., ... & Chang, K. J. (2021). Benefit, tolerance, and safety of hybrid argon plasma coagulation for treatment of Barrett's esophagus: US pilot study. Endoscopy International Open, 9(12), E1870-E1876.

Knabe, M., Beyna, T., Rösch, T., Bergman, J., Manner, H., May, A., ... & Ehlken, H. (2022). Hybrid APC in Combination With Resection for the Endoscopic Treatment of Neoplastic Barrett's Esophagus: A Prospective, Multicenter Study. The American journal of gastroenterology, 117(1), 110-119.

The author’s answer: Limited attention had been given to Hybrid-APC. The research progress of Hybrid-APC has now been incorporated into the discussion section of the article.

11. In your overall conclusion I am missing a statement regarding surgery as a therapeutic option.

The author’s answer: Thank you for your suggestion. Given that only one of the studies we analyzed involved surgery as an intervention, we approached making comparisons between surgery and other treatment options in the Discussion section with caution. It is undeniable that endoscopic treatments maintain the normal esophageal anatomy, are safer, and are linked to lower morbidity and mortality rates compared to esophagectomy. For these reasons, endoscopic interventions are increasingly becoming the primary therapy for patients with high-grade dysplasia (HGD) or intramucosal cancer, replacing surgery.

Reviewer #4:

1. In the "study selection", the article divided the treatments into 10 major groups. I wonder if it could have been possible to group similar treatments together to create fewer groups.

The author’s answer: Thank you for your valuable feedback, which is greatly appreciated. Prior to this, we had made attempts to categorize similar interventions, such as grouping proton pump inhibitors and H2 receptor antagonists together. Further consolidation could potentially impact the accuracy of the research findings.

2. In the "study selection", it was mentioned that studies with small sample sizes were excluded. It would be helpful to specify the criteria used to determine what is considered a small sample size.

The author’s answer: Thanks for your suggestion, we define small sample size studies as those with less than 20. We have added it in the article and marked it in red.

3. While there seems to be a fundamental difference in the meaning of "complete ablation of BE" and "complete eradication of dysplasia," in practical terms, they appear to have very similar criteria. It would be beneficial to mention the rationale for separating these two, as they seem quite similar in clinical application.

The author’s answer: Thanks for your suggestion, we have added it to Study Selection of the article and marked it in red.

4. In Figure 1, the number of cases classified as "irrelevant" is quite high (805 cases). It would be helpful if the specific criteria used to make this determination were explicitly stated.

The author’s answer: Thank you for your comment. We define irrelevant studies as those that are not related to the treatment of Barrett's esophagus.

5. Generally, dysplasia and esophageal adenocarcinoma are assessed using different criteria. If the article refers to "complete ablation of BE" and places adenocarcinoma on the same level as dysplasia, it would be useful to explain the reasoning behind this.

The author’s answer: When it comes to the complete ablation of Barrett's esophagus, we combined adenocarcinoma and dysplasia into a single group for statistical analysis, given the limited number of studies and the small sample size specifically related to adenocarcinoma.

6. If the article suggests that adenocarcinoma can progress directly from metaplasia without going through dysplasia, it might be more helpful to mention this in the introduction or study design section rather than just in the discussion. This could help reduce confusion.

The author’s answer: We would like to express our gratitude for your time and the valuable opportunity you have provided us to enhance the manuscript. Our study has indicated that depending solely on endoscopic surveillance or PPI/H2RA treatment alone might not be sufficient for managing BE. This conclusion is supported by the research of SHARMA P et al., who noted that a subset of patients with intestinal metaplasia could progress rapidly to invasive carcinoma without the usual progression through dysplasia. These insights have been integrated into the discussion section of the article.

---

## [Decision Letter · Decision Letter 1]

1 Apr 2024

Comparison of interventions for Barrett's esophagus: a network meta-analysis

PONE-D-23-16418R1

Dear Dr. Pan,

We’re pleased to inform you that your manuscript has been judged scientifically suitable for publication and will be formally accepted for publication once it meets all outstanding technical requirements.

Kind regards,

Alejandro Piscoya

Academic Editor

PLOS ONE

Additional Editor Comments (optional):

Reviewers' comments:

Reviewer's Responses to Questions

**Comments to the Author**

1. If the authors have adequately addressed your comments raised in a previous round of review and you feel that this manuscript is now acceptable for publication, you may indicate that here to bypass the “Comments to the Author” section, enter your conflict of interest statement in the “Confidential to Editor” section, and submit your "Accept" recommendation.

Reviewer #1: All comments have been addressed

Reviewer #3: All comments have been addressed

2. Is the manuscript technically sound, and do the data support the conclusions?

Reviewer #1: Yes

Reviewer #3: Yes

3. Has the statistical analysis been performed appropriately and rigorously? 

Reviewer #1: Yes

Reviewer #3: Yes

4. Have the authors made all data underlying the findings in their manuscript fully available?

Reviewer #1: Yes

Reviewer #3: Yes

5. Is the manuscript presented in an intelligible fashion and written in standard English?

Reviewer #1: Yes

Reviewer #3: Yes

6. Review Comments to the Author

Reviewer #1: Barrett's esophagus (BE) is a precancerous condition which can develop into esophageal cancer (EC) without correct management. For now, the standard surveillance for BE is endoscopy regularly based on the grade dysplasia of BE in order to detect early carcinoma. The widely accepted treatment to BE is endoscopic eradication therapy,including APC, EMR and ESD. Radiofrequency ablation (RFA) is another option eradication of BE. Photodynamic therapy (PDT)is a rare chioce for BE. However, considering the high preveilence of esophageal squamous cell carcinoma,PDT was frequently used in China for some time past, and the Chinese researchers have more experiences in PDT. I think we can give the authors a chance to provide a different view for management of BE although their manuscript based on uncommon strategy for BE.

Reviewer #3: Thanks for including MPEC in the treatment groups, but you forgot to also included it in the abbreviations section.

7. PLOS authors have the option to publish the peer review history of their article (what does this mean?). If published, this will include your full peer review and any attached files.

Reviewer #1: No

Reviewer #3: No

---

## [Editor Report · Acceptance letter]

26 Apr 2024

PONE-D-23-16418R1 

PLOS ONE

Dear Dr. Pan, 

I'm pleased to inform you that your manuscript has been deemed suitable for publication in PLOS ONE. Congratulations! Your manuscript is now being handed over to our production team.

Kind regards, 

on behalf of

Professor Alejandro Piscoya 

Academic Editor

PLOS ONE